# HYPERQUERY: A FRAMEWORK FOR HIGHER ORDER LINK PREDICTION

## ABSTRACT

Groups with complex set intersection relations are a natural way to model a wide array of data, from the formation of social groups to the complex protein interactions which form the basis of biological life. While graphs are a natural way to represent complex networks and are well studied, typical approaches to modeling group membership using graphs are lossy. Moreover, a simple graph based approach cannot be used for prediction and classification over a collection of entities. Hypergraphs are a more natural way to represent such "higher order" relationships, but efforts to apply machine learning techniques to hypergraph structured datasets have been limited thus far. In this paper, we address the problem of link prediction in knowledge hypergraphs as well as simple hypergraphs and develop a novel, simple, and effective optimization architecture to solve this task. Additionally, we study how integrating data from node-level labels can improve the results of our system. Our self-supervised approach achieves significant improvement over state of the art results on several hyperedge prediction and knowledge hypergraph completion benchmarks.

## 1 INTRODUCTION

There is a significant demand for applying learning to graph structured data over the past couple of years. While graphs can accurately model binary relations between entities, they are not a natural representation of n-ary relations between entities. For example, a protein complex network cannot be represented by a graph since a protein complex might be created only in a presence of more than two proteins Giurgiu et al. (2019). In this paper, we set out to answer complex queries that go beyond a simple graph i.e. the current graph learning algorithms cannot solve such problems without major modifications. In particular, we study *Learning on Hypergraphs*.

Hypergraphs are a generalization of graphs for representing such n-ary relations. Formally, a hypergraph $H$ is a tuple $(V, E)$ where $V$ is a set of *nodes*; $E \subseteq 2^{|V|}$ is a set of nonempty subsets of $V$ called *hyperedges*. Similarly, a *knowledge hypergraph* is a generalization of a knowledge graph where relations are between any number of entities. Recent research shows that hypergraph models produce more accurate results even in problems in which graphs are used to represent n-ary relations Zhou et al. (2006); Feng et al. (2020); Fatemi et al. (2019).

In this paper, we aim to solve the task of *hyperedge prediction* on both simple and knowledge hypergraphs. Hyperedge prediction in simple hypergraphs is analogous to link prediction in graphs, and can be formally defined as follows: *given a hypergraph $H = (V, E)$ and a k-tuple of nodes $(v_1, v_2, ..., v_k)$, predict whether this tuple forms a hyperedge or not*. While link prediction in graphs is a well-studied problem, hyperedge prediction has not received adequate attention in spite of its many applications. For example, it can be used to predict new protein complexes, drug-drug interactions, new collaborations in citation networks, discover new chemical reactions in metabolic networks, etc. Yadati et al. (2020); Giurgiu et al. (2019); Piñero et al. (2019). Hyperedge prediction is a more challenging problem than link prediction in graphs. Formulating this problem as a link prediction problem in graphs is a lossy operation, reducing the accuracy of predictions Kirkland (2017); Tu et al. (2018).

In knowledge hypergraphs, it is often necessary to not only predict new hyperedges but also their type. For example, in a protein-drug genomics knowledge hypergraph, it is important to predict

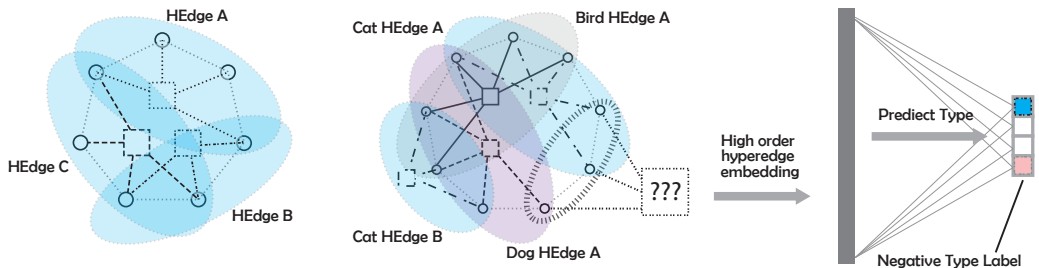

Figure 1: The HyperQuery Inference Problem: In this paper, we prefer to work with the *star-expansion* representation of a hypergraph (*left*). We are interested in studying hypergraphs and knowledge hypergraphs with categorical labels (types) stored on the edges (*center*). Our primary objective is to create a system that predicts the existence of a hyperedge and its type.

not only drug-drug interactions but also the type of these interactions that describes the side effects Zitnik et al. (2018). This generalized hyperedge prediction problem can be described formally as follows: *given a knowledge hypergraph $KH = (\mathcal{E}, \mathcal{R})$ and a tuple of entities $(e_1, e_2, ..., e_k)$, we want to predict if this tuple forms a hyperedge and if so, what its type is*. This problem setting is illustrated in figure 1.

This paper makes the following contributions.

- **HyperQuery:** We describe *HyperQuery*, a neural message passing based framework that is designed to find embeddings of hyperedges in a semi-supervised fashion.
- **Novel feature extraction:** We use clustering to extract global features of nodes and hyperedges.
- **Higher order link prediction:** We solve hyperedge prediction on simple hypergraphs as well as knowledge hypergraphs.

The pipeline of our system is shown in figure 4.

## 2 RELATED WORK

**Link prediction in graphs:** There are two ways to address the link prediction problem in networks. One approach is to learn an embedding of the nodes of a graph and then apply a function on these embeddings to obtain the embedding of an edge. We call this approach an *indirect* approach to solve hyperedge prediction. For example, node2vec Grover & Leskovec (2016), and deepwalk Perozzi et al. (2014) are random-walk based approaches and Vashishth et al. (2019); Davidson et al. (2018) are GNN-based approaches for graphs that learn the embedding of the nodes and then use a binary operator such as average to compute the embedding of an edge i.e. given two nodes $u$, and $v$, they apply binary operator $o$ to generate an embedding of $g(u, v)$ such that $g : V \times V \longrightarrow \mathcal{R}^d$ where $d$ is the dimension of the embedding. *Direct* approaches in graphs learn the embedding of edges directly and apply that to the link prediction problem. Examples of this approach includes path-based approaches such as Zhu et al. (2021); Sadeghian et al. (2019). One popular task in knowledge graphs is to predict missing relations between entities. This task is called knowledge graph completion and one could think of it as a generalization of link prediction in graphs. The problem of knowledge graph completion has been studied extensively for example: Zhu et al. (2021); Wang et al. (2020); Rossi et al. (2022).

**Hyperedge prediction:** Link prediction in hypergraphs can also be done indirectly by first computing the embedding of the nodes of the hypergraph and then applying a function $g$ to the tuple $(v_1, v_2, ..., v_k)$ to obtain the embedding of the tuple i.e. hyperedge. The difficulty with such models is that for hypergraphs, function $g$ must be nonlinear to capture higher-order proximity of nodes in the hypergraph Tu et al. (2018). This means it is not a good design choice to use an operator such as average to obtain the embedding of a hyperedge from its nodes. Related works HyperSAGNN and NHP have used non-linear functions such graph neural networks Zhang et al. (2020); Yadati et al. (2020) to compute the embedding of a hyperedge from its nodes. Directly learning the embedding of

a hyperedge and applying hyperedge prediction to that embedding has not been studied. To the best of our knowledge, we are the first to approach the problem in this manner.

Link prediction in knowledge hypergraphs is relatively under explored. Most approaches generalizes knowledge graph based methods to work with n-ary relations. More recent work on this area propose novel methods that directly works with knowledge hypergraphs such as Fatemi et al. (2019); Wang et al. (2021).

**Random Walk Based Feature Generation for hypergraphs:** In problems in which nodes do not have features, the first step is to generate features for the nodes. The most common approach is to use node2vec. For example, HyperSAGNN Zhang et al. (2020) first generates features using node2vec and then passes these features through an attention layer. During inference, the embedding of a proposed tuple of nodes is computed using a one layer, position-wise feed-forward network. They can also use the corresponding row of the adjacency matrix to extract features. Similarly, NHP runs node2vec on the clique expansion of a hypergraph and then uses a GCN layer to improve these features. Finally, they pass these embeddings to a scoring layer Yadati et al. (2020).

**Clustering Based Feature Generation:** Random-walk based approaches are one way to generate features for a hypergraph, but they only exploit the local connectivity of nodes in the hypergraph. In this paper, we investigate the effectiveness of hypergraph clustering algorithms for feature extraction.

Clustering has been used to understand the structure of complex networks. A cluster in a hypergraph is a partition of nodes into sets that are similar to each other and dissimilar from the rest of the hypergraph. Intuitively, a cluster is a group where nodes are densely inter-connected and sparsely connected to other parts of the network. Placing the embedding of such nodes closely in the embedding space can be useful in many graph mining tasks such as clustering, node classification, network reconstruction and link prediction Bhowmick et al. (2020). A classic clustering algorithm in hypergraphs is the *FM* Fiduccia & Mattheyses (1982) algorithm: given an initial assignment of nodes to clusters, this algorithm moves a node to the cluster that results in the largest reduction in connectivity between clusters. Multi-level clustering approaches such as Karypis et al. (1999); Devine et al. (2006); Maleki et al. (2021) build on this algorithm and improve the performance of FM algorithm by successively coarsening the hypergraph, finding clusters in the smallest hypergraph, and then interpolating these to the coarser hypergraphs, applying the FM algorithm at each level. In this paper, we use a recent work called BiPart, Maleki et al. (2021), for clustering to generate initial features of the hyperedges as well as nodes of a hypergraph.

## 3 HYPEREDGE CONVOLUTION OPERATOR

We begin with the question of how one might build a system to "answer" HyperQueries. Given a collection of nodes, our HyperQuery oracle will answer questions such as whether these nodes are related. Ultimately, our goal is to perform prediction and classification on a collection of nodes.

In this section, we start by studying the question of inference in knowledge hypergraphs, where a vector of data is stored on entities and relations. We present an approach for defining trainable convolutions on hyperedges which is well suited for performing inference for knowledge hypergraph completion. In the next section, we will discuss the pre-processing approach that completes our framework for performing hyperedge prediction on simple hypergraphs.

A basic approach for defining a HyperEdge convolution operator proceeds in two steps. First, we set out to aggregate and summarize the known label data in the neighborhood of the hyperedge $e$ or query set of node $v$ in question. Then, we apply learned weight matrix $W^k$ and pass it through the non linearity function $\sigma$.

$$h_e^k = \sigma(W^k \cdot \Omega\{h_v^{k-1} \forall v \in \mathcal{N}(e)\}) \tag{1}$$

To fully unlock the power of modern machine learning, we set out to create a trainable system. In fact, we improve on Equation 1 and propose something more general. A flexible, composable, trainable,

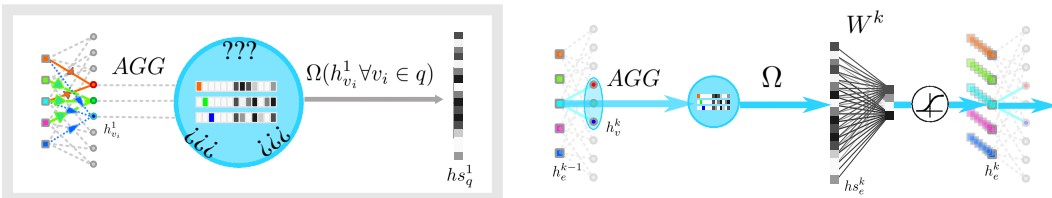

Figure 2: HyperEdge Convolution Operator. A *HyperQuery* can be passed by specifying a set of nodes $q$ into the system (*left*). For each node, we first aggregate over its neighboring hyperedges to update its embedding, then for each hyperedge we perform *summary function* $\Omega$, to compute what we call the *harmonic statistic* of $HQ$. (*right*) By adding a fully connected layer $W^k$ and a nonlinearity $\sigma$ in series with this calculation, *we arrive* at our proposed HyperEdge convolution operator.

hyperedge message-passing based (i.e. scalable) operation for learning to answer HyperQueries:

$$h_v^k = AGG(h_e^{k-1}, \forall e \in \mathcal{N}(v)) \tag{2}$$

$$hs_e^k = \Omega(\{h_v^k \forall v \in \mathcal{N}(e)\}) \tag{3}$$

$$h_e^k = \sigma(W^k \cdot hs_e^k) \tag{4}$$

For each node $v$ in hyperedge $e$, we aggregate along it's hyperedge neighbor features to calculate $h_v$. Then, for each hyperedge $e$, we aggregate the messages from its nodes (*i.e.*, $h_v \forall v \in \mathcal{N}(e)$), using a *summarization* function $\Omega(.)$ as shown above. Finally, we have a hidden "deconvolution" layer and a non-linearity. The $h_{v_i}^k$ above amounts to a "mean-field" embedding of each node $v_i$ as the average of a sampling of the data stored on the hyperedges it is connected to.

This concludes the first part of our approach to learning useful representations of HyperQueries. Intuitively, this message passing based aggregation and summarization process can be thought of a *diffusion process* or a *flow*, which averages and symmetrizes the edge level data by effectively performing a step of Laplacian smoothing.

In a knowledge hypergraph, we can effectively learn the hidden weights $W^k$ by choosing the initial feature $h_e^0$ of a hyperedge $h_e$ to be the one hot vector of its type, and building up an autoencoder style problem. By repeating this iteration twice, we already arrive at an effective tool for knowledge hypergraph completion which generalizes to unseen data far better than benchmark approaches. In Table 6, we outline our performance relative to other benchmarks. We achieve this performance with a straightforward model, we simply treat the output $h_e^2$ as a predicted class label $\mathcal{P}(e)$. From there we minimize the loss below as a function of hidden layer weights $W^1$ and $W^2$.

$$\mathcal{L} = \sum_{e \in \mathcal{D}} \mathcal{J}(\mathcal{P}(e), \mathcal{G}(e)) \tag{5}$$

In our objective function, we take $\mathcal{J}(.)$ to be the cross entropy loss, $e$ to be a hyperedge in the training dataset $\mathcal{D}$, $\mathcal{P}(e)$ is the predicted type of hyperedge $e$, and $\mathcal{G}(e)$ is the actual ground truth type (relation) of hyperedge $e$ in the dataset.

## 4 ANSWERING HYPERQUERIES ON SIMPLE HYPERGRAPHS

Our main focus in this paper is the task of *hyperedge prediction*, i.e. given an *simple* hypergraph and set of nodes $(v_1, .., v_n)$, predict whether this tuple forms a hyperedge.

In this setting, there is no additional label data such as knowledge hypergraphs and thus inference must be computed entirely from the hypergraph structure. Thus if we want to make use of a flow based approach, we need a way of computing a useful initialization. Our insight is that useful initial categorical embeddings can be found by existing clustering algorithms, which works by recursively finding balanced cuts of a hypergraph.

Figure 3: Generating Useful Labels using clustering: Not all hypergraph data comes with labels (*left*), but so far our approach has depended on having categorical labels on the edges and nodes. We leverage hypergraph clustering to first assign cluster id as labels to the nodes, obtaining an initial cluster assignment $\ell_v^0$ (*center*), which we then propagate to the hyperedges via Max Pooling (*right*) in order to obtain the edge cluster assignment $\ell_e^0$. We use $\ell_e^0$ as the data in $\mathcal{L}(h)$.

## 4.1 FEATURE EXTRACTION USING CLUSTERING

One way to learn an embedding that captures meaningful structural information about a hypergraph is to run random-walk based algorithms such as node2vec Grover & Leskovec (2016). Previous works on hyperedge prediction such as HyperSAGNN Zhang et al. (2020) and NHP Yadati et al. (2020) use node2vec to first learn feature vectors for nodes of a hypergraph represented in a variety of ways. Then, they improve these features using attention-based methods. However, such methods have a number of drawbacks:

1. They do not learn the correlation between nodes that do not appear in the same random walk, might require many samples to capture effects of weak correlations.
2. Random walks explore only the local properties of the hypergraph, optimization might get stuck in local minima.
3. Random walks on hypergraphs mix very quickly and thus even if an algorithm converges to a good answer, will likely require many samples/computation to converge.
4. The published methods learn node based features and they do not learn feature vector of hyperedges directly, couldn't be directly applied to our architecture.

It occurred to us that instead, it is possible to use state-of-the-art partitioning based clustering tools as an alternative approach for pre-computing features. While clustering is a well studied problem in ML, for instance it has been used in prediction as early as in 2015 Deylami & Asadpour (2015), it is not used by any of the state-of-the-art prediction tools today, and therefore, it merits revisiting, given the simplicity and effectiveness of the approach we evaluate in this paper.

In particular, for our HyperQuery flow, we want an initial label assignment that is "smooth" in a way that is well defined for the given hypergraph topology. To do this, we run a clustering algorithm that partitions the nodes of the hypergraph into clusters. Each cluster of nodes is given a unique integer id. To transfer these labels onto hyperedges, we perform a max-pooling step (described in more detail in Figure 3).

**Cut-Metric for Hypergraphs.** For our purpose, we can use any clustering algorithm. In our experiments, we use a multilevel clustering algorithm that partitions nodes into a given number of clusters while minimizing the hyperedge cut, defined below. While there is not a unique canonical way to define a hyper-edge cut, this approach worked well and exposes the number of clusters as a hyper-parameter to the system.

$$cut(H, C) = \sum_e (\lambda_e(H, C) - 1) \qquad (6)$$

Here, $H$ is the hypergraph, $C$ is the partition of nodes into clusters, and $\lambda_e(H, C)$ is the number of clusters that hyperedge $e$ spans. Intuitively, nodes that belong to the same cluster are considered similar. This is similar to approaches like node2vec in which nodes that appear in the same random walk are considered to be similar.

## 5 OPTIMIZING THE HYPEREDGE CONVOLUTION OPERATOR

Here we outline a few important technical details that significantly improve our framework.

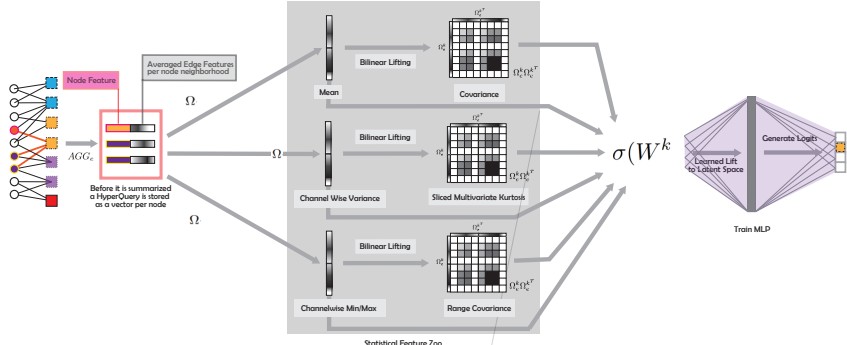

Figure 4: Learning $W^1$. Given a hypergraph with labels on nodes and edges, we experiment with a zoo of choices for $\Omega$. In this illustration, we convolve once, multiply by $W^1$ and $\sigma$ and then pass through a fully connected layer to return to a space of the same dimension of hyperedge types. Finally we then train as an autoencoder.

## 5.1 NODE CLUSTER AS A FEATURE

The initial aggregation step $AGG$ in our convolution operator comes from the idea that a reasonable first way to think about creating embeddings for hyperedges is to simply average node embeddings. However, this simple measure fails to capture important information regarding the distribution of node embeddings within a hyperedge.

To fix this problem, we augment every node feature vector with the a one-hot encoding of the node cluster id, as seen in Equation 7:

$$h_v^k = CONCAT\{AGG\{h_e^{k-1}, \forall e \in \mathcal{N}(v)\}, x_v\} \tag{7}$$

where $AGG(.)$ is an aggregation function, and $x_v$ is the one-hot partition id of a node. If nodes of a hypergraph have features themselves, these features can also be concatenated with $x_v$.

## 5.2 CHOOSING THE RIGHT $\Omega$

In this work, we evaluate three different types of summary statistics for $\Omega$ (Figure 4, center). A component-wise mean and variance, and a sort of "robust variance" estimator we call *minmax* (element-wise difference of maximum and the minimum values of the vectors) similar to the one used in NHP Yadati et al. (2020) to achieve state of the art performance. Intuitively, when $\Omega$ is the mean, convolution maps hyperedges to the average embedding of constituent nodes, whereas the variance measures how correlated different nodes are with each other. In our experiments in Section 6, we find that for the hyperedge prediction, *minmax/variance* performed better as a choice of $\Omega$ whereas in knowledge hypergraph completion, *mean* worked better.

We also augment these aggregations to better capture the correlation between different labels/communities by introducing a bilinear aggregation layer Zhu & Koniusz (2021). Bilinear aggregation has significantly improved visual concepts recognition Koniusz et al. (2017), and in practice also generally improves the accuracy of our results. Specifically, *we create an auto-correlation matrix by multiplying a hyperedge feature vector by itself*. This matrix has dimension $d \times d$, which we flatten.

## 5.3 AUGMENTED CONVOLUTION OPERATOR

Our final scheme for hyperedge message passing is the following:

$$\Omega_e^k = \Omega\{h_v^k, \forall v \in \mathcal{N}(e)\} \tag{8}$$

$$h_e^k = \sigma(W^k \cdot flat(\Omega_e^k \Omega_e^{k^T})) \tag{9}$$

Where $h_v^k$ is computed using Equation 7, $\Omega$ is an aggregation function, $flat(.)$ flattens the correlation matrix and stores it in a column vector, $AGG(.)$ takes the element-wise mean of the hyperedge

feature vector, $x_v$ is the feature vector of node $v$, and $W^k$ is the trainable weight matrix. The effect of using different aggregation functions $\Omega$ is explored empirically in Table 4.

**Time complexity.** We analyze the time complexity of our framework (Equations 7, 8,and 9) in terms of the size of the hypergraph, assuming the number of relations (types) is independent of the size of the hypergraph. Let $m$ denote the number of hyperedges and $n$ denote the number of nodes. Let $deg(v_i)$ denote the degree of node $v_i$ and let $\tilde{\Delta}$ denote $max_{1 \leq i \leq n} deg(v_i)$. For Equation 7, our framework takes $O(n \cdot \tilde{\Delta})$ time. For Equation 9, let $deg(e_i)$ denote the degree of hyperedge $e_i$ and let $\Delta$ denote $max_{1 \leq i \leq m} deg(e_i)$. Equation 8 takes $O(m \cdot \Delta)$ time. Finally, the time complexity of Equation 9 is constant since the number of labels is independent of the size of the graph. Overall, our framework takes $O(m \cdot \Delta) + O(n \cdot \tilde{\Delta})$.

## 5.4 MODEL EXPLAINABILITY

The ideal initial feature vector for a hyperedge should include local and global properties of the hypergraph. In our model, clustering exposes global properties. Specifically, we keep track of the cluster id of hyperedges as well as the distribution of labels on the nodes in the neighborhood of a hyperedge. Intuitively, Hyperedges that are in the same cluster or have similar label distribution on their nodes are similar to each other and by assumption, they are related. Previous work Grover & Leskovec (2016); Perozzi et al. (2014) has shown that placing such similar nodes closely in the embedding space will facilitate tasks such as node classification and link prediction. We follow the same intuition.

The message-passing scheme of labels is conceptually similar to label propagation Raghavan et al. (2007). The objective of the label propagation algorithm is to assign each node into a cluster with the most number of its neighbouring nodes. Intuitively, this scheme also put hyperedges with the same label/cluster closer in the embedding space facilitating hyperedge prediction for both knowledge hypergraphs and simple hypergraphs.

## 6 EXPERIMENTS

We evaluate our framework HyperQuery on knowledge hypergraph completion task as well as hyperedge prediction and show that our architecture outperforms the state of the arts for these tasks. Furthermore, we conduct an ablation study to understand the effectiveness of the major HyperQuery kernels. The details of our setting are in Appendix A.

### 6.1 KNOWLEDGE HYPERGRAPH COMPLETION

We first evaluate our system for the task of link prediction for knowledge hyperedges i.e. knowledge hypergraph completion. We evaluate our system on three datasets: FB-AUTO, M-FB15K, and JF17K. We used the same test, train, and validation sets as Fatemi et al. (2019). We use MRR (mean reciprocal rank) and Hit@1, 3 (hit ratio with cut-off values of 1 and 3) as our evaluation metrics.

**Datasets:** The datasets used in this section are standard knowledge hypergraph datasets from previous work Fatemi et al. (2019). The summary of the knowledge hypergraphs are in Table 1. FB-AUTO, M-FB15K Fatemi et al. (2019), and JF17K Wen et al. (2016) are knowledge hypergraphs with n-ary relations that is collected from Freebase dataset.

Table 1: Knowledge hypergraph dataset

| DATA SET | $|\mathcal{E}|$ | $|\mathcal{R}|$ | #train | #valid | #test |
|---|---|---|---|---|---|
| FB-AUTO | 3,410 | 8 | 6,778 | 2,255 | 2,180 |
| M-FB15K | 10,314 | 71 | 415,375 | 39,348 | 38,797 |
| JF17K | 29,177 | 327 | 77,733 | 15,822 | 24,915 |

We compare our results with the following baselines: **HSimplE, and HypE** two embedding based approaches for knowledge hypergraphs introduced by Fatemi et al. (2019). Both methods find the

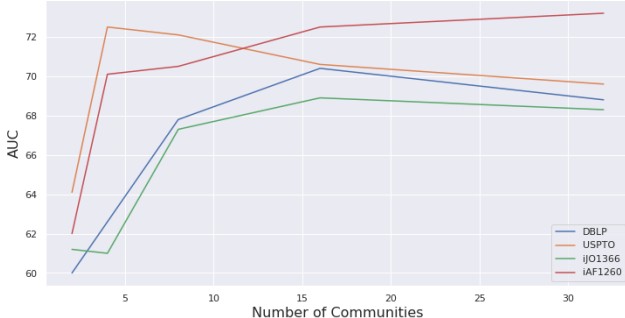

Figure 5: Performance of HyperQuery for different numbers of clusters.

embedding of an entity based on its position in a relation. **M-DistMult** Yang et al. (2014) defines a scoring function for each tuple $(e_1, r_1, e_2)$. We modify this so that each relation could have any number of entities. **M-TransH** is a modified standard knowledge graph scoring function (TranshH) that accepts beyond binary relations. Finally, **M-CP** Hitchcock (1927) is a tensor decomposition approach. We apply a similar approach as Fatemi et al. (2019) to extend it beyond binary relations.

We used HyperQuery to perform knowledge hypergraph completion. Table6 summarizes the result of our experiment. HyperQuery performs the best compared to all other baselines on all metrics. Specifically, on metric Hit@1 against the best baseline, it improves up to $5\%$ on dataset FB-AUTO, $18\%$ on M-FB15K, and more than $50\%$ on JF17K.

Table 2: Knowledge Hyperedge Completion.

| | **FB-AUTO** | | | **M-FB15K** | | | **JF17K** | | |
| | MRR | *Hit@1* | *Hit@3* | MRR | *Hit@1* | *Hit@3* | MRR | *Hit@1* | *Hit@3* |
|---|---|---|---|---|---|---|---|---|---|
| HYPERQUERY | **91.5** | **83.1** | **99.9** | **95.1** | **90.5** | **99.7** | **99.9** | **99.1** | **99.9** |
| HSIMPLE | 79.8 | 76.6 | 82.1 | 73.0 | 66.4 | 76.3 | 47.2 | 37.8 | 52.0 |
| HYPE | 80.4 | 77.4 | 82.3 | 77.7 | 72.5 | 80.0 | 49.4 | 40.8 | 53.8 |
| M-DISTMULT | 78.4 | 74.5 | 81.5 | 70.5 | 63.3 | 74.0 | 46.3 | 37.2 | 51.0 |
| M-TRANSH | 72.8 | 72.7 | 72.8 | 62.3 | 53.1 | 66.9 | 44.4 | 37.0 | 47.5 |
| M-CP | 75.2 | 70.4 | 78.5 | 68.0 | 60.5 | 71.5 | 39.1 | 29.8 | 44.3 |

## 6.2 HYPEREDGE PREDICTION

The second task we study in this paper is hyperedge prediction. We evaluate our system on four datasets: iAF1260b, iJO1366, USPTO, and DBLP. We used $70\%$ of the hyperedges in these for test, $10\%$ for validation and $20\%$ for training. The summary of these datasets are in Table 3.

Table 3: Hyperedge prediction dataset

| DATA SET | NODES | HYPEREDGES | TYPE OF DATA |
|---|---|---|---|
| IAF1260B | 1,668 | 2,084 | METABOLIC REACTIONS |
| IJO1366 | 1,805 | 2,253 | METABOLIC REACTIONS |
| USPTO | 16,293 | 11,433 | ORGANIC REACTIONS |
| DBLP | 20,685 | 30,956 | CO-AUTHORSHIP |

**iAF1260b**[1] a metabolic reaction dataset for specie E. coli. We use this dataset for hyperedge prediction where our goal is to predict missing reactions i.e. hypereges. In this dataset each reaction is considered as a hyperedge connecting its participating metabolites (nodes).

**iJO1366** [1] a metabolic reaction dataset similar to iAF1260b. **USPTO** [2] a organic reaction dataset. We used a subset of chemical substances that only contains carbon, hydrogen, nitrogen, oxygen,

phosporous, and sulphur. **DBLP** [3] a co-authorship publication dataset. We used a subset of papers published in only AI conferences: AAAI, IJCAI, NeurIPS, ICML, CVPR, ICCV, ACL, NAACL, etc. Each author in this dataset is a node and papers represent hyperedges connecting authors of a paper.

We compare our results with previous works: NHP Yadati et al. (2020), HyperSAGNN Zhang et al. (2020), HyperGCN Yadati et al. (2019), and node2vec Grover & Leskovec (2016). The description of these methods are in 2. We use a similar negative sampling strategy as NHP Yadati et al. (2020). Motivated by NHP, we investigate another aggregation function $\Omega$ for our approach. We call this aggregation function *minmax* which is the element wise difference of max and min values of the embedding vectors in equation 7. Furthermore, we also experiment on using operator *variance* as an aggregation function. These results is summarized in Table 4. We use 16 communities for this experiment. In Section 6.3, we discuss the effect of different number of communities.

HyperQuery with minmax aggregation function performs the best on all datasets in this paper. Comparing with NHP, on their minmax operator, HyperQuery minmax outperform them by up to 7% on iAF1260b, 5% on iJO1366, 1.5% on USPTO, and 1.8% on DBLP; on mean aggregation operator, HyperQuery outperforms NHP by up to 6% on iAF1260b, 4.6% on iJO1366, 12% on USPTO, and 13.6% on DBLP. HyperQuery outperforms HyperSAGNN on all datasets by more than 6%, and HyperGCN by more than 4%. Finally, node2vec results show poor performance on hypergraph datasets showing that graph based approaches are not suitable for hypergraphs.

## 6.3 ABLATION STUDY

We study the behaviour of each component of HyperQuery. First, we study the effect of bilinear pooling on hyperedge prediction. Table 4, shows this effect. On the larger dataset, USPTO, the bilinear pooling improves the quality of our framework by 5.6% for minmax. Similarly, for DBLP, the improvement is about 2%. On the smaller datasets, bilinear pooling does not make a significant change.

Next, we study the effect of the number of clusters in our dataset. Figure 5 shows how the performance of our framework changes as we increase or decrease the number of clusters. For all datasets except iJO1366, the performance improves as we increase the number of clusters up to 16 clusters and then decreases. This behaviour is expected since as we increase the number of clusters, the quality of these clusters decreases (*i.e.*, the hyperedge cut increases). However, if the number of clusters is very small, we are not exploring the global structure of our hypergraph enough which again decreases the performance of our framework.

Table 4: Area Under Curve (AUC) scores for hyperedge prediction. NOPOOL refers to the model without bilinear pooling. HQ is our proposed model.

|  | IAF1260B AUC | IJO1366 AUC | USPTO AUC | DBLP AUC |
|---|---|---|---|---|
| HQ-MINMAX | 72.2 | 68.5 | **75.7** | **72.0** |
| HQ-MEAN | 66.5 | 65.9 | 72.2 | 69.6 |
| HQ-VAR | 71.1 | 68.1 | 75.0 | 71.3 |
| HQ-MINMAX-NOPOOL | **72.7** | **68.9** | 70.1 | 70.4 |
| HQ-MEAN-NOPOOL | 67.6 | 65.7 | 70.6 | 65.7 |
| HQ-VAR-NOPOOL | 72.3 | 67.5 | 74.1 | 71.6 |
| NHP-MINMAX | 64.3 | 63.2 | 74.2 | 69.2 |
| NHP-MEAN | 60.5 | 61.2 | 65.5 | 56.4 |
| HYPER-SAGNN | 60.1 | 56.3 | 67.1 | 65.2 |
| HYPERGCN-MINMAX | 64.0 | 62.2 | 70.5 | 67.4 |
| NODE2VEC-MINMAX | 66.0 | 62.0 | 71.0 | 67.0 |

---

[1] `https://github.com/muhanzhang/HyperLinkPrediction`
[2] `https://github.com/wengong-jin/nips17-rexgen`
[3] `https://github.com/muhanzhang/HyperLinkPrediction`

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

## A  DETAILS ON HYPERQUERY SETTINGS

**Negative Sampling.** We use a similar approach to Yadati et al. (2020) i.e for each hyperedge $e$ in our dataset, we create a hyperedge $e'$ by having half of the vertices sampled from $e$ the remaining half from $V - e$. This sampling method is motivated by the chemical reaction datasets where it is unlikely that half of the substances of a valid reaction (from $e$) and randomly sampled substances (from $V - e$) are involved in another valid reaction.

**Hyper-parameters.** The number of clusters used in the evaluation section of this paper is 16. Furthermore, we showed the effect of using different number of clusters in Figure 5. We use 2 layers of hyperedge convolution in our test implementation.

**Operator $\Omega$.** for a given set of vectors $x_1, ..., x_k \in \mathcal{R}^d$,

$$MINMAX(x_1, ..., x_k) = (max\ x_{sl} - min\ x_{sl}), s \in [k], l = 1, ..., d \tag{10}$$

$$VAR(x_1, ..., x_k) = (variance(x_{sl})), s \in [k], l = 1, ..., d \tag{11}$$

$$MEAN(x_1, ..., x_k) = (MEAN(x_{sl})), s \in [k], l = 1, ..., d \tag{12}$$

## B  CLUSTERING ALGORITHM

In this paper we use BiPart Maleki et al. (2021) as our clustering algorithm. BiPart is a multilevel and deterministic hypergraph partitioner. Given a number $k$, BiPart partitions the nodes of the hypergraph into k disjoint blocks. In our framework, we use BiPart as a pre-processing step where we partition the nodes of a hypergraph first and the we use the one hot vector of their partition id as the initial feature of the nodes of the hypergraph. Finally, these initial features are used as an input to the HyperQuery framework.

In practice, any hypergraph partitioning algorithm can be used to partition the nodes of the hypergraph.

## C  HYPEREDGE CLASSIFICATION

One neutral choice for our system would be to use it to solve hyperedge classification. In such problems, a hypergraph $H$ is a tuple $(V, E, \mathcal{L})$ where $V$ is a set of *nodes*; $E \subseteq 2^{|V|}$ is a set of nonempty subsets of $V$ called *hyperedges*; and $\mathcal{L}$ is a set of labels for hyperedges. Our task is given a hypergraph and labels on a small subset of hyperedges, predict labels on the remaining hyperedges.

We evaluate our system on three datasets: Cora, Citeseer, and Pubmed 5. We used $20\%$ of the hyperedges in these datasets for test, $10\%$ for validation and $70\%$ for training (for 20 epochs). Since there are no previous approaches for hyperedge classification, we modified existing methods as baselines for our problem:

**HyperNetVec Maleki et al. (2022):** an unsupervised multi-level approach to generate the representation of a hypergraph. This method uses an existing embedding system to generate an initial

embedding and further improve these embeddings using a refinement algorithm. This methods generates embeddings for nodes of a hypergraph. We modified this method to also generate embeddings for hyperedges.

**node2vec(mean):** a random walk based approach to generate representation of a graph in a semi-supervised manner. In order to use node2vec, we convert our hypergraph to a graph and then generate embeddings for the nodes of the hypergraph. We obtain embedding of a hyperedge by performing a mean aggregation on the embedding of its nodes.

We used HyperQuery to run these experiments. Table 6 summarizes the result of our experiment. HyperQuery performs the best compared to HyperNetVec and node2vec. It improves up to $2.3\%$ on dataset Cora, $18\%$ on Citeseer, and $4\%$ on Pubmed.

Table 5: Real world hypergraph dataset for classification

| DATA SET | NODES | HYPEREDGES | TYPE OF DATA | CLASSES |
|---|---|---|---|---|
| CORA | 2,709 | 1,963 | CITATION | 7 |
| CITESEER | 3,328 | 2,182 | CO-AUTHORSHIP | 6 |
| PUBMED | 19,717 | 12,971 | CO-AUTHORSHIP | 3 |

Table 6: Hyperedge classification. Accuracy in $\%$ and time in seconds.

| | CORA | | CITESEER | | PUBMED | |
|---|---|---|---|---|---|---|
| | ACCURACY | TIME | ACCURACY | TIME | ACCURACY | TIME |
| HYPERQUERY-MEAN | **78.5** | 10 | **75.3** | 12 | **82.4** | 25 |
| HYPERNETVEC | 76.2 | 28 | 57.1 | 45 | 78.1 | 33 |
| NODE2VEC-MEAN | 76.0 | 30 | 57.1 | 31 | 74.2 | 140 |

