# OpenReview forum: "HyperQuery: A Framework for Higher Order Link Prediction"
_ICLR.cc/2023/Conference — Submitted to ICLR 2023_

### Official Review · Reviewer_McmT · 2022-10-20

**Confidence:** 4
**Correctness:** 3
**Technical Novelty And Significance:** 3
**Empirical Novelty And Significance:** 3
**Recommendation:** 6

**Clarity, Quality, Novelty And Reproducibility:**

The paper is novel. It is a new framework that combines local and global characteristics for higher-order link prediction. With respect to quality, as with every framework, it is a combination of multiple methods. So, it seems fine, and no problem can be observed. However, there are important problems with respect to clarity and reproducibility.

The clarity must be improved before publication. Even though it seems a simple framework based on the figures, it is very complicated to understand the process. Unfortunately, this also affects the reproducibility of the paper (even though the codes are available). Just as an example, the clustering process is based on "state-of-the-art hierarchical partitioning based clustering tools", but the final algorithm is not mentioned in the paper. Similarly, the selection of the number of clusters is supposed to be based on equation 7, but again, it is difficult to figure out the selection given that almost no description is given. Similar problems can be observed in sections 3 and 5.

The previous problems also affect the reproducibility of the experiments. For example, no details are given about the MLP and the number of clusters for each problem. What is the clustering algorithm (hierarchical is just mentioned)? What are the characteristics of the MLP? How do you select the number of clusters?


**Strength And Weaknesses:**

The introduction is well-written and stated the main problem. The related work covers several points of topics and papers. The methodology of the experiment section is also strong. It has several baselines, different problems are considered, and an ablation study is applied to check the importance of each stage of the framework. However, there are several weaknesses that must be improved.

The most important issue is the clarity of the paper. The paper tries to be general, so different approaches can be considered. This can be observed with the use of functions such as AGG, \Omega, and others. However, this generality loses the main focus of the paper, and the framework becomes complex (even though it is simple). So, in general, the three sections explaining the main framework must be improved.

The use of space can also be improved. For example, most of subsection 4.1 can be resumed in a single paragraph. Figure 2 can be explained in a single paragraph, and figure 1 from the appendix could be incorporated into the paper. You can also add the time complexity of the framework (a small part is mentioned in the appendix). This will make the paper stronger.

Another problem is reproducibility. Unfortunately, several details are omitted. From the use of the clustering algorithm, to all the hyperparameters used in the framework.

Minor comments:
Figure 1: it says project instead of paper.
Figure 1 ("right" part): it says Prediect
Figure 5: use a white background, increase the size of the font for the legend, and improve the caption mentioning that it is for a specific experiment.


**Summary Of The Paper:**

The paper proposes a new framework for higher-order link prediction. The paper creates hyperedges embedding (local information) and combines this information with clustering embedding (global information). Both pieces of information were supposed to be combined using an MLP for the final classification process.


**Summary Of The Review:**

The paper is well-written and proposes a new interesting framework. The results are very interesting, and the work could be publishable. However, the current state of the paper must be improved in terms of clarity and reproducibility.

---

> ### Author Response · Authors · 2022-11-15
> **Additional comments**
>
> Thank you for your careful suggestions and feedback. We appreciate your time and review. We have submitted a new draft of the paper that we believe addresses your concerns. Here is our attempt to answer your questions:
>
> *The most important issue is the clarity of the paper.*
> - We improved the clarity of the paper in the new draft.
>
> *Figure 2 can be explained in a single paragraph, and figure 1 from the appendix could be incorporated into the paper. You can also add the time complexity of the framework*
> - Thank you for your suggestions. We incorporated them in our new draft.
>
> *Several details are omitted. From the use of the clustering algorithm to all the hyper-parameters used in the framework.*
> - We added a section in the appendix to explain the hyper-parameters in more detail which we hope will improve reproducibility.

---

### Official Review · Reviewer_t9m9 · 2022-10-22

**Confidence:** 3
**Correctness:** 3
**Technical Novelty And Significance:** 2
**Empirical Novelty And Significance:** 2
**Recommendation:** 5

**Clarity, Quality, Novelty And Reproducibility:**

The clarity in both methodology and evaluation needs some improvement. The technical contribution and novelty is marginally significant.

**Strength And Weaknesses:**

Strength
1.The authors study an interesting and practical problem to generalize edge classification to hyperedge with many real-world applications.
2.The proposed method is a natural generalization to GNN from graph to hyper-graph with alternating node and edge representation learning.
3.The authors carry out experiments on several datasets with comparison to existing state-of-art methods.

Weakness
1.The authors should provide more details on the empirical evaluation. First, how important hyper-parameters are set like the number of iteration of message-passing k. Also it would be better to have an experiment on the sensitivity of the parameters. Second, for hyper-edge classification, what is the evaluation protocol e.g. how are the negative samples generated and what is positive to negative ratio etc. In Table 4, does HQ the same as HyperQuery. Also, what is the exact definition of the minimax operation and the bilinear operation used?
2.Besides hyper-graph based approaches, the authors should compare the proposed method to graph-based methods as well. A neutral choice would be casting the problem as a graph classification problem by flattening all hyper-edges. Comparison to some simple heuristics like graph structure features like number of edges would also be very helpful.
3.The authors mention the classification of hyperedge types. Seems no experiment is carried out in section 6? Also in section 6.2, why use 70% for test and only use 20% for training?


**Summary Of The Paper:**

In this paper, the authors study the problem of hyperedge classification in hypergraphs. The authors generalize the idea of GNN to carry out message-passing type methods to alternate between edge and node representations. Experiments are carried out on several datasets with comparison to existing hyperedge classification methods.

**Summary Of The Review:**

The authors studied an interesting problem with a natural generalization of GNN to hypergraph as solution. However, both clarity and empirical needs some improvements.

---

> ### Author Response · Authors · 2022-11-15
> **Additional comments**
>
> Thank you for your careful review. We tried to address most of your concerns in the new draft.
> Here is our answer to some of your questions:
>
> *The authors should provide more details on the empirical evaluation*
> - We added a section in the appendix explaining the hyper-parameters as well as details on the empirical evaluation.
>
> *how are the negative samples generated*
> - We mentioned in the paper that we use a similar approach as NHP [1] for generating negative samples. For each hyperedge e in our dataset, we create a hyperedge e′ by having half of the vertices sampled from e the remaining half from V − e. This sampling method is motivated by the chemical reaction datasets.  We added a more detailed explanation in the appendix.
>
> *HQ the same as HyperQuery*
> - Yes, we fixed this in the draft.
>
> *what is the exact definition of the minimax*
> - the element-wise difference of maximum and the minimum values of the vectors. We explained this in more details in the main paper and appendix.
>
> *A neutral choice would be casting the problem as a graph classification problem by flattening all hyper-edges*
> - We have added these results to the appendix and compared it with graph based approaches.
>
> *What is the definition of Bilinear*
> - We create an auto-correlation matrix by multiplying a hyperedge feature vector by itself
>
> *the authors should compare the proposed method to graph-based methods as well*
> - Various approaches have been tried in the literature, and we ran our code on standard benchmarks.  We also show a few graph-based approaches such as node2vec in this paper. Furthermore, If you look at the baseline methods we compare to in table 4.  In those works, they further compare against earlier approaches as well as graph-based approaches.  Moreover, in the hypergraph literature, it has been shown that graph-based approaches generally perform worse
>
> *The authors mention the classification of hyperedge types. Seems no experiment is carried out in section 6?*
> - These are the results we document in Table 2.  We significantly outperform prior approaches on this benchmark.
>
> *Also in section 6.2, why use 70% for test and only use 20% for training?*
> - To have a fair comparison, we are following the same setting as previous works. Therefore, we are using the same test/train set for our experiments.

---

### Official Review · Reviewer_dmzc · 2022-10-24

**Confidence:** 3
**Correctness:** 3
**Technical Novelty And Significance:** 3
**Empirical Novelty And Significance:** Not applicable
**Recommendation:** 5

**Clarity, Quality, Novelty And Reproducibility:**

The quality and clarity of the writing needs to be improved. Th originality is good.

**Strength And Weaknesses:**

Strength：
1. The method in this paper shows consistent superiority on experiments with multiple data sets.
2. The authors identified the shortcomings of their previous work and innovatively improved it through simple methods.

Weaknesses：
1. The abstract and the introduction section does not state the background of the task, the motivation of the method and where the innovation lies. Instead, the authors define the hypergraph, knowledge hypergraph, and hypergraph prediction tasks in this section, which I think should be explained in the model section.
2. The captions for Figures 2 and 3 are too long. The description of the model should be placed in the body of the text.
3. The composition of the pipeline should be further rethought to allow the reader to better understand the model details.
4. What are equations 1 and 2 for? They are not used in the subsequent objective function Eq. 6.
5. The logic of the article is not clear, making it difficult for the reader to catch the point. The authors should rethink the organization of their paper to better explain their work.

**Summary Of The Paper:**

The model first generates natural labels by a clustering algorithm and then performs hyperedge prediction by hypergraph convolution.

**Summary Of The Review:**

This work identified the shortcomings of the previous work and improved it using simple methods. But the writing is not really satisfactory. The author should revise it significantly to make the work more acceptable to the readers.

---

> ### Author Response · Authors · 2022-11-15
> **Additional comments**
>
> Thank you for your suggestions and careful review. Your suggestions have been incorporated in the new draft of the paper. Please see the new draft as we improved it upon your suggestions.
>
> *The abstract and the introduction section does not state the background of the task, the motivation of the method and where the innovation lies.*
> - We improved this in the new draft.
>
> *The captions for Figures 2 and 3 are too long.*
> - They have been shortened in the new draft
>
> *The composition of the pipeline should be further rethought to allow the reader to better understand the model details.*
> - We have simplified our discussion of the convolution operator, and combined it with a discussion of the choice of \Omega. We hope the new version we uploaded is clearer.
>
> *What are equations 1 and 2 for? They are not used in the subsequent objective function Eq. 6.*
> - Thank you for this suggestion.  We removed these equations and it significantly simplified the paper.
>
> *The logic of the article is not clear, making it difficult for the reader to catch the point. The authors should rethink the organization of their paper to better explain their work.*
> - Point taken.  We hope the edited version we just uploaded allays your main concerns.  We are prepared to further polish the writing and figures ahead of the camera ready deadline.

---

### Official Review · Reviewer_Fuim · 2022-10-29

**Confidence:** 3
**Correctness:** 3
**Technical Novelty And Significance:** 2
**Empirical Novelty And Significance:** 2
**Recommendation:** 3

**Clarity, Quality, Novelty And Reproducibility:**

- Clarity: Low. Unlike most papers that are poorly presented, this paper is mostly free of grammatical errors and typos, but the descriptions of key components in the text and figures are either missing details or unclear.
- Quality: Medium. Proposed framework and experimental evaluation appear to be technically sound, but I can't say that with high confidence due to the poor presentation.
- Novelty: Medium. There appear to be some novel components in the proposed HyperQuery framework, but the main novelty is in addressing the hyperedge prediction problem, where there is not much existing literature.
- Reproducibility: Medium. Code is provided, although I did not examine it in detail. There are not enough details in the experiment descriptions and supplements for me to reproduce the experiments. I'm not even certain of how the evaluation is being conducted, as indicated by my questions above.


**Strength And Weaknesses:**

Strengths:
- Considers the task of hyperedge prediction, which is more difficult and less well studied than link prediction in simple graphs.
- Improvements in accuracy on both prediction tasks is impressive compared to prior methods, including recently proposed neural hyperedge prediction methods HyperSAGNN and NHP.

Weaknesses:
- Presentation is unclear in many respects. Although the paper is mostly free from grammatical errors, the text and figures are confusing in many aspects. See my questions below.
- Different elements of the proposed HyperQuery framework seem to be chosen and combined in ad-hoc manner that doesn't provide much insight into why they work.
- Novelty of different elements in the HyperQuery framework is unclear.

Questions:
1. What is Figure 2 supposed to show? On the left pane, I can't tell what the hyperedges are. What are the colored dashed lines supposed to show? Are these the hops in the random walk? What is the right pane supposed to show?
2. What are the quantities \#train, \#valid, and \#test referring to? They don't appear to map to either the number of entities or relations. Are these counts of hyperedges?
3. I don't understand the evaluation in Section 6.1. What counts as a hit for hit\@1 and hit\@3? Is there a way to give partial credit if HyperQuery is given a query for a subset of entities that have a hyperedge? For example, if a hyperedge exists between $(e_1, e_2, e_3, e_4)$ but the query is for $(e_1, e_2, e_3)$, what is the correct answer, and how is it evaluated?

Typos and minor issues:
- Figure 1: Prediect -> Predict
- Section 5.2, first line: varience -> variance
- Placement of Figure 2 on page 3 seems odd. It is not referenced anywhere around that page, and not until Section 4.1 on page 5. Perhaps Figure 2 should be moved after Figure 3, which is referenced much earlier. Similarly, Figure 5 is in a strange location, showing up before Tables 2-4 despite not being referred to until afterwards.
- Inconsistent use of the notation $\mathcal{N}(\cdot)$. $\mathcal{N}(v_i)$ is stated to denote the neighborhood of node $v_i$ (incorrectly denoted as $N(v_i)$ in the Figure 3 caption). Then what is $\mathcal{N}(e)$ for a hyperedge $e$? Is it a union of the neighborhoods for all nodes $v_i \in e$?

**Summary Of The Paper:**

The authors propose the HyperQuery framework for predicting hyperedges between nodes in hypergraphs. It can also be used to predicted hyperedge types, e.g., in a knowledge hypergraph. Link prediction in hypergraphs is much less studied and more difficult than link prediction in simple graphs. The proposed approach defines an embedding for each hyperedge. These are then aggregated to compute an embedding for each node. They use hypergraph clustering to initialize the embeddings. The authors demonstrate improved empirical results for both knowledge hypergraph completion and hyperedge prediction on several real data sets.

**Summary Of The Review:**

This paper addresses an important task, hyperedge prediction in hypergraphs, that has not been very well studied previously. The proposed HyperQuery framework appears sound and may have some novel components, but I am left confused about many elements of this paper. It is quite strange to me that the text in the paper is grammatically sound but lacks clarity in descriptions. Similarly, the figures look good, but I am not sure what several of them are supposed to show.

I think there is a lot of potential in this paper but would not recommend it for publication at this time. If the presentation can be significantly improved, it could become a high quality paper in the future.

---

> ### Author Response · Authors · 2022-11-15
> **Additional Clarification**
>
> Thank you for your review. We believe we have addressed most of your concerns in the new draft.
>
> *Presentation is unclear*
>  - Based on the feedback from reviewers, we have improved the presentation and submitted a new draft.
>
> *Different elements of the framework seem to be chosen and combined in ad-hoc manner*
> - Could you please elaborate more on what elements of the framework you are referring to? Specifically, we added extra figures (i.e. fig 2,3) to further explain the intuition behind our model. We also included an ablation study to further explain the importance of each element of our framework. Furthermore, the different choices of \Omega within the convolution operator are drawn from best practices similar to those we found in other works, and in Table 4 you can see that on our test sets, the choice of aggregation operator makes a significant impact on system performance.
>
> *Novelty of different elements in the HyperQuery framework is unclear.*
>
> - There are two main novelties in our proposed framework, which we sought to highlight in figures 2 and 3 of the original submission.  The first key idea is that we introduce a novel (to our knowledge) formulation of a hyperedge convolution operator.  The second key idea is that we convolve categorical data, like edge labels, or communities generated by a hierarchical partitioning algorithm, using this operator.  Using categorical features hasn’t been tried for hyperedge prediction before and our implementation consistently outperforms competing baselines, sometimes by a rather large margin.
>
> *What is Figure 2 supposed to show*
> - Since this figure was causing confusion, we removed it from the main paper. The point of this figure was to show random walks on hypergraphs mix very quickly and even if an algorithm based on node2vec features converges to a good answer eventually, it will likely require many samples to converge in practice.
>
> *What are the quantities #train, #valid, and #test referring to*
> - We have used the exact same settings as the previous work [1] (as mentioned in the paper) for these experiments. These counts the number of hyperedges and you can think of relations as the type/label of hypereges.
>
> - [1] Naganand Yadati, Vikram Nitin, Madhav Nimishakavi, Prateek Yadav, Anand Louis, and Parth Talukdar. Nhp: Neural hypergraph link prediction., CIKM ’20,
>
> *What counts as a hit for hit@1 and hit@3? Is there a way to give partial credit if HyperQuery is given a query for a subset of entities that have a hyperedge*
> - Hits@N denotes the proportion of the tuples whose target entities are ranked within top N. The partial credit is more meaningful and could be considered for the task of link prediction for simple hypergraphs. For the task of predicting the relations, it is not very useful.
>
> *Typos and minor issues*
> - We have fixed those in the main paper. Thank you.

---

> > ### Comment · Reviewer_Fuim · 2022-12-02
> > **Thanks for the clarifications**
> >
> > Thanks for the clarifications. After reading through your comments and the other reviews, I didn't see much to change my overall opinion of the paper.
> >
> > I do think the proposed idea has potential and could make for a nice paper in the future, if heavily revised. The main weakness I identified was the presentation quality.

---

### Author Response · Authors · 2022-11-15
**Cover letter**

We would like to thank all the reviewers for their careful reviews, insightful feedback, and detailed suggestions. There seems to be a consensus among the reviewers that the organization of the paper could be improved. We have been working on the paper and incorporated your suggestions in the new draft of the paper. We are submitting a new draft with the following changes:
- The organization has been improved
- Captions of the figures have been decreased
- Figure 2 is removed
- We moved figure 4 and the time complexity section from the appendix to the main paper
- A more detailed description of the hyper-parameters has been added to the appendix

---

### Decision · Program_Chairs · 2023-01-20

**Decision:**

Reject

**Justification For Why Not Higher Score:**

The paper looks like a practice submission by student without careful proofreading. It is far from the average presentation level of ICLR papers.

**Justification For Why Not Lower Score:**

N/A

**Metareview: Summary, Strengths And Weaknesses:**

This paper proposes a hyperedge prediction framework. Messages are passed between nodes and hyperedges alternatively and final hyperedge representations are pooled from node embeddings. The main weakness of the paper is that the idea of treating hypergraph as bipartite graph and performing message passing between hyperedges and nodes is not new. Many previous works have by default used this scheme. The presentation in this paper is also bad. There are many typos and notation issues (e.g., the AGG() never appears in Eq.9). The figures are generally too small. The citation format is wrong, with many in-text citations that should be replaced by parenthesis citations. All the above make the paper hard to read. Also, many used datasets miss citing or discussing their original papers.